# Serum Sclerostin Level Is Negatively Associated with Bone Mineral Density in Hemodialysis Patients

**DOI:** 10.3390/medicina58030385

**Published:** 2022-03-04

**Authors:** Jing-Wun Lu, Ru-Jiang Syu, Chih-Hsien Wang, Bang-Gee Hsu, Jen-Pi Tsai

**Affiliations:** 1Department of Physical Medicine and Rehabilitation, Hualien Tzu Chi Hospital, Buddhist Tzu Chi Medical Foundation, Hualien 97004, Taiwan; jingwunlu@gmail.com; 2Division of Nephrology, Department of Internal Medicine, Dalin Tzu Chi Hospital, Buddhist Tzu Chi Medical Foundation, Chiayi 62247, Taiwan; jamrickson@gmail.com; 3Division of Nephrology, Hualien Tzu Chi Hospital, Buddhist Tzu Chi Medical Foundation, Hualien 97004, Taiwan; wangch33@gmail.com; 4School of Medicine, Tzu Chi University, Hualien 97004, Taiwan

**Keywords:** bone mineral density, sclerostin, Dickkopf-1, hemodialysis

## Abstract

*Background and Objectives*: Sclerostin and Dickkopf-1 (DKK1) modulate osteoblastogenesis, but their role in bone loss in hemodialysis (HD) patients is inconclusive. This study investigated relationships among lumbar bone mineral density (BMD), serum sclerostin, and DKK1 in HD patients. *Materials and Methods*: Blood samples were obtained from 75 HD patients. Dual-energy X-ray absorptiometry measured lumbar BMD of the lumbar vertebrae (L2–L4). Enzyme-linked immunosorbent assay revealed serum sclerostin and DKK1 concentrations. *Results*: There were 10 (13.3%), 20 (26.7%), and 45 (60%) patients defined as presenting with osteoporosis, osteopenia, or normal BMD, respectively. Age, alkaline phosphatase, urea reduction rate, fractional clearance index for urea, sclerostin level, and percentage of female patients are significantly negatively associated with the lumbar BMD and T-score, while the body mass index and waist circumference significantly positively associated with the lumbar BMD and T-score. Multivariate forward stepwise linear regression analysis indicated that serum sclerostin (β = −0.546, adjusted R2 change = 0.454; *p* < 0.001), age (β = −0.216, adjusted R2 change = 0.041; *p* = 0.007), and percentage of female HD patients (β = −0.288, adjusted R2 change = 0.072; *p* = 0.0018) were significantly negatively associated with lumbar BMD in HD patients. *Conclusions*: Advanced age, female gender, and serum sclerostin level, but not DKK1, were negatively associated with BMD in HD patients.

## 1. Introduction

The growing rate of chronic kidney disease (CKD), along with comorbidities, place a heavy burden on the general health and medical resources of society [1]. CKD is associated with a higher risk of fractures due to dysregulated bone metabolism and lower bone mineral density (BMD) as renal function worsens, which results in a substantially poor prognosis compared to that of the general population [2,3,4]. In addition, evidence showed that end-stage renal disease on dialysis exhibited a higher prevalence of fractures than pre-dialysis CKD patients [3]. In hemodialysis (HD) patients, abnormal bone turnover presented as low bone mass and density was common, and 9.5–23% were defined as osteoporosis and 16.7–45% as osteopenia based on BMD [5,6]. Additionally, an increased risk of fractures led to a major public health burden worldwide [7].

Risk factors associated with osteoporosis in CKD were multifactorial, including poor nutrition, vitamin D deficiency, hyperparathyroidism, metabolic acidosis, limited physical activity, and CKD-related metabolic mineral bone disease [8]. In addition, two recently and intensively studied factors known as sclerostin and Dickkopf-1 (DKK1), which are osteocyte- or osteoblast-derived regulators that affect osteoblast bone formation and vascular calcification through Wnt/β-catenin signalling, were associated with osteoporosis and fractures in CKD patients [9,10,11]. The serum sclerostin level increased in conjunction with a reduction of BMD in diabetes mellitus (DM) patients [12], and the values of sclerostin increased as renal function worsened. A negative relationship was found with BMD in HD patients [13,14]. In addition, the baseline sclerostin levels could predict bone loss in HD patients [11]. Nevertheless, a puzzling positive relationship was found between sclerostin and BMD in post-menopausal women [15], pre-dialysis CKD [16], peritoneal dialysis (PD) [17], and HD patients [18,19]. Regarding DKK1, studies showed that it was neither related to BMD nor to serologic biomarkers in HD patients [11,18]. In addition, DKK1 demonstrated an inverse relationship in pre-dialysis CKD patients [16].

Since the roles of sclerostin and DKK1 in mineral bone disorders remain inconclusive and the relationship between serum sclerostin and DKK1 with BMD in HD patients is controversial, the aim of this study was to examine the association of sclerostin and DKK1 with lumbar BMD in HD patients.

## 2. Materials and Methods

### 2.1. Patients

The patients who were enrolled in this study were older than 50 years of age and were receiving standard 4-h dialysis 3 times per week using standard bicarbonate dialysate and a high-flux polysulfone disposable artificial kidney (FX class dialyzer, Fresenius Medical Care, Bad Homburg, Germany) for at least 3 months from June 2015 to August 2015 at a medical center in Hualien, Taiwan. Patients were excluded if they received treatment for osteoporosis (bisphosphonates, teriparatide, or estrogen medications); presented with a history of lumbar fracture or surgery, acute infection, malignancy, acute myocardial infarction, pulmonary edema, or heart failure at the time of blood sampling, or if they declined to provide informed consent. The participants enrolled in the control group were age-, gender, and DEXA-matched to HD patients and history of diabetes mellitus and hypertension were reviewed by medical records or usage of oral hypoglycemic or anti-hypertensive medications. The estimated glomerular filtration rate was calculated by CKD-EPI (Chronic Kidney Disease Epidemiology Collaboration) formula. This study was approved by the Research Ethics Committee of Hualien Tzu Chi Hospital of the Buddhist Tzu Chi Medical Foundation (Approval No.: IRB106-62-B, Approval date: 5 June 2017).

### 2.2. Anthropometric Analysis

Before receiving HD, body weight (BW), and body height (BH) were measured to the nearest half-kilogram and half-centimeter, and waist circumference (WC) was measured to the nearest half-centimeter at the shortest point below the lower rib margin and the iliac crest. Body mass index (BMI) was calculated as (BW)/(BH)^2^ (kg)/(m)^2^ [20,21].

### 2.3. Biochemical Investigations

Blood samples of control participants were collected in the morning after 8-h fasting. Before receiving HD therapy, fasting blood samples (~5 mL) of HD patients were collected and immediately centrifuged at 3000× *g* for 10 min, stored at 4 °C, and analyzed within 1 h after collection. Serum values of the biochemical variables were measured using an autoanalyzer (Siemens Advia 1800, Siemens Healthcare GmbH, Henkestr, Germany). The adequacy of HD was calculated using the fractional clearance index for urea (Kt/V) and urea reduction ratio (URR) based on a formal and single-compartment dialysis urea kinetic model. Values of serum sclerostin and DKK1 (Biomedica Immunoassays, Vienna, Austria) and intact parathyroid hormone (iPTH) (IBL International GmbH, Hamburg, Germany) were examined by commercially enzyme-linked immunosorbent assays (ELISA) [21].

### 2.4. Bone Mineral Density Measurements

After blood sampling, patients were immediately given BMD measurements before HD. Lumbar vertebrate (L2–L4) BMD was measured using dual-energy X-ray absorptiometry (DEXA; QDR 4500, Hologic Inc., Marlborough, MA, USA) and expressed as an absolute value (g/cm^2^) T-score (deviation from peak BMD) [20]. The T-score was defined as the number of standard deviations (SD) from the mean BMD of gender-matched young control subjects. Compared to the control value, a lumbar bone T-score less than −2.5 was used as the diagnostic cutoff for osteoporosis, and a lumbar bone T-score from −1.0 to −2.5 was used for the diagnosis of osteopenia, which was based on the criteria established by the World Health Organization [22].

### 2.5. Statistical Analysis

The Kolmogorov–Smirnov test was used to assess the normal distribution of clinical variables. According to the distribution of data (normal distribution or skewed), continuous variables were expressed as mean ± SD or as medians and interquartile ranges. Meanwhile, differences among groups (normal, osteopenia, and osteoporosis) were examined with the Kruskal–Wallis test or one-way analysis of variance, and expressed as median with a 1st to 3rd interquartile range or mean ± SD, accordingly. Categorical variables were expressed as the number of patients and were analyzed using the *χ^2^* test. Skewed distributions were log-transformed for recalculation by normal distribution. Correlation between clinical variables and lumbar BMD as well as serum sclerostin were evaluated using univariable and multivariate forward stepwise linear regression analysis for independent variables. All statistical analyses were performed using SPSS for Windows (version 19.0; SPSS Inc., Chicago, IL, USA). A *p*-value < 0.05 was considered to be statistically significant.

## 3. Results

In this study, there were 75 HD and 65 control participants enrolled. Of the control group, there were 40 and 25 participants with estimated GFR more than 60 m/mL/1.73 m^2^ and 45–60 mL/min/1.73 m^2^, respectively. Compared to control group, HD patients had lower total calcium but higher waist circumference, BUN, Creatinine, phosphorus, iPTH, sclerostin (153.32 (92.11–207.50) pmol/L vs. 63.37 (40.52–83.27) pmol/L, *p* < 0.001) and DKK1 (13.63 (7.42–21.87) pmol/L vs. 7.00 (4.88–10.03) pmol/L, *p* < 0.001) (Table 1).

Of the HD patients, 45 (60%), 20 (26.7%), and 10 (13.3%) were in the normal, osteopenia, and osteoporosis group, respectively (Table 2). Compared to the normal group, HD patients in the osteopenia or osteoporosis groups were older (*p* = 0.008); consisted of more females (*p* = 0.001); of higher values of sclerostin (*p* < 0.001), alkaline phosphatase (ALP) (*p* = 0.025), URR (*p* = 0.006), and Kt/V (*p* = 0.008); exhibited a lower BMI (*p* = 0.002). Values of DKK1 or iPTH showed no significant differences among these three groups.

Simple linear analysis revealed that BH (*r* = 0.495, *p* < 0.001), BW (*r* = 0.545, *p* < 0.001), BMI (*r* = 0.406, *p* = 0.003), WC (*r* = 0.397, *p* < 0.001), and serum creatinine (*r* = 0.277, *p* = 0.016) were positively correlated with lumbar BMD, while female gender (*r* = −0.337, *p* = 0.003), age (*r* = −0.337, *p* = 0.003), serum ALP level (*r* = −0.278, *p* = 0.016), URR (*r* = −0.366, *p* = 0.001), Kt/V (*r* = −0.347, *p* = 0.002), and sclerostin (*p* = −0.679, *p* < 0.001) were negatively correlated with lumbar BMD (Table 3). Multivariable stepwise linear regression analysis of variables associated with lumbar BMD in univariate linear regression analysis (female, age, body height, BW, WC, body mass index, creatinine, alkaline phosphatase, URR, Kt/V, and sclerostin) revealed that serum sclerostin (β = −0.546, adjusted *R^2^* change = 0.454; *p* < 0.001), age (β = −0.216, adjusted *R^2^* change = 0.041; *p* = 0.007), and female gender (β = −0.288, adjusted *R^2^* change = 0.072; *p* = 0.001) were significantly and independently negatively associated with lumbar BMD.

Figure 1 shows significantly higher BW (Figure 1A, 71.38 ± 14.0 kg vs. 56.12 ± 11.71 kg, *p* < 0.0001) and BMI (Figure 1B, 26.34 ± 4.83 kg/m^2^ vs. 23.26 ± 4.44 kg/m^2^, *p* = 0.005) but lower sclerostin (Figure 1C, 128.92 (81.14–168.44) pmol/L vs. 177.86 (104.0–256.17) pmol/L, *p* = 0.008) of male HD patients than female HD patients. The levels of sclerostin significantly increased as eGFR decreased, and the HD patients had the highest levels (Figure 1D).

Simple linear analysis revealed that lumbar BMD (*r* = −0.679, *p* < 0.001), lumbar T-score (*r* = −0.687, *p* < 0.001), BH (*r* = −0.443, *p* < 0.001), BW (*r* = −0.491, *p* < 0.001), BMI (*r* = −0.382, *p* = 0.001), and WC (*r* = −0.394, *p* < 0.001) were negatively correlated with serum sclerostin, while female gender (*r* = 0.331, *p* = 0.004), serum ALP level (*r* = 0.238, *p* = 0.039), URR (*r* = 0.346, *p* = 0.002), and Kt/V (*r* = 0.329, *p* = 0.004) were positively correlated with serum sclerostin (Table 4). Multivariable stepwise linear regression analysis of variables associated with serum sclerostin in univariate linear regression analysis (female, BH, BW, WC, BMI, alkaline phosphatase, URR, Kt/V, and lumbar BMD, T-score) revealed that lumbar BMD (β = −0.687, adjusted *R^2^* change = 0.465; *p* < 0.001) was significantly and independently negatively associated with serum sclerostin.

## 4. Discussion

The major findings of this study indicated that advanced age and female gender were associated with lower lumbar BMD, while the serum sclerostin level but not DKK1 was negatively associated with lumbar BMD in HD patients. This association remained even after adjustments were made for clinical variables.

For patients with impaired renal function, evidence indicated a higher prevalence of CKD-related mineral bone disease, including dysregulated mineral metabolism, metastatic calcification, and abnormal quantity and quality of bone, which led to a higher risk of fracture and poor long-term survival [6,23,24]. Therefore, the 2017 Kidney Disease: Improving Global Outcomes (KDIGO) guidelines recommended dual-exergy X-ray (DEXA) for evaluating BMD to assess fracture risks in this population [23]. Because the lower BMD associated with lower renal function leads to a higher risk of osteoporosis and fracture-related mortality compared to that of the general population, studies exist that investigate the potential risk factors associated with a lower BMD [11,25,26]. In a cohort study of that used DEXA to assess pre-dialysis CKD patients consisting of 33% osteopenia cases and 8% osteoporosis cases, risk factors associated with lower BMD were female, advanced aged, lower BMI, and progressive decline of renal function [25]. A cross-sectional study revealed that advanced CKD patients older than 65 years and menopause-stage females were significantly more osteoporotic [26]. Moreover, DEXA examination of dialysis patients revealed higher prevalence of osteoporosis at the spine and hip, and both spine and hip were associated with lower BMD, advanced age, female gender, and bone-specific ALP [11]. Similarly, 26.7% and 13.3% of HD patients respectively defined as osteopenia or osteoporosis demonstrated that BMD was negatively associated with age and female gender.

After being discovered as a 22-kD osteocyte-derived bone morphogenetic protein, sclerostin, along with DKK1, which was secreted by osteoblasts and osteocytes as a 26-kD small glycoprotein, could bind to low-density lipoprotein receptor-related protein 5 and 6 co-receptors to modulate bone formation by impeding the Wnt signaling pathway [27,28]. The loss of sclerostin can lead to increased bone formation, as was revealed by extensive studies on the role of sclerostin in vascular calcification and bone formation in CKD patients [9,10,11,16,29]. Evidence revealed that sclerostin was higher as the BMD lower in DM patients [12], and studies revealed that as renal function worsened, higher levels of sclerostin were present, which highlighted its role in the pathogenesis and high prevalence of osteoporosis in CKD patients [2,3,4,14,18]. From the cross-sectional study, higher serum sclerostin was correlated with markedly lower renal function and lower BMD in children with CKD who were on regular HD [13]. In this study, we similarly found that sclerostin correlated negatively with renal function and lumbar BMD, while HD patients had the highest levels of sclerostin compared to those of control participants. Of adult HD patients, the serum sclerostin levels were higher than in patients without CKD and was markedly correlated with bone turnover markers in the adjusted analysis, which indicated that sclerostin could be a promising non-invasive biomarker for high turnover bone disease [30]. Furthermore, the baseline sclerostin level could independently predict bone loss in the total hip and an increase in the sclerostin level, which was correlated with trabecular bone loss of the spine of patients on dialysis [11]. The role of sclerostin was investigated by intervention trials using an anti-sclerostin antibody in post-menopausal women with low BMD, which substantially increased spine and hip BMD by 17.7% and 6.2%, respectively, compared to baseline levels [31] with a lower incidence of vertebral fractures [32], while a meta-analysis showed a marked increase in spine, hip, and femoral neck BMD compared to those of anti-resorptive therapies, anabolic agents, or placebos [33]. However, contradicting our current knowledge of sclerostin functions, some studies unexpectedly found that a positive correlation was found between the serum sclerostin level and BMD in post-menopausal women [15] and pre-dialysis and dialysis patients [16,17,18]. One possible assumption could be that the production of sclerostin was a direct reflection of higher body weights and BMD in men, thus reflecting a higher total body skeletal mass and greater number and activity of osteocytes [16,17,18], while another speculative explanation could be that high sclerostin might act as a Wnt antagonist, as reflected by an increase in the Wnt/β-catenin signaling pathway [15]. Our results did not support these theories, as in our study, male HD patients had significantly higher BW and BMI than female HD patients, but male HD patients had less levels of sclerostin than female HD patients. In addition, there was a significantly negative correlation between sclerostin and body weight, BMI, and BMD, independent of gender. Therefore, the results of this study were in accordance with previous studies showing that there was a negative association between sclerostin and BMD and echoed the studies using agents antagonizing the effects of sclerostin to increased BMD and to prevent osteoporosis in post-menopausal women [31,32,33]. Taken together, we showed that serum sclerostin elevated as renal function decreased and especially correlated negatively with body weight, BMI, and lumbar BMD in HD patients.

Regarding the relationship between DKK1 and BMD, studies disclosed inconsistent results. Some studies showed that DKK1 demonstrated a negative correlation with BMD in pre-dialysis CKD and DM patients [12,16], but this relationship was not observed in post-menopausal women or in HD and PD patients [15,17,18,30], and a decrease in BMD could not be predicted in HD patients [11]. Although sclerostin and DKK1 are negative regulators of the Wnt/β-catenin signaling pathway and either could individually contribute to the regulation of bone architecture, the wider distribution of the production of DKK1 [16] and no significant correlation between DKK1 and BMD in this study led us to assume that the serum DKK1 level might not be a valuable biomarker for predicting bone status in HD patients.

The limitations of this study were, first, that it was cross-sectional with a limited number of HD patients. Second, we did not collect the data of physical activities of HD patients, which could influence BMD as well as the production of sclerostin. Third, only lumbar spine BMD was measured, but the correlation between sclerostin and BMD at the other sites were not investigated. Fourth, we record only comorbidities including DM and HTN but did not record medications nor measured serum vitamin D level of HD patients in this study. Therefore, before a cause–effect relationship can be established, using the co-activity of serum sclerostin and DKK1 to predict the BMD of HD patients should be confirmed by further longitudinal studies.

## 5. Conclusions

Along with age and female gender, a negative correlation was found between the lumbar BMD and serum sclerostin but not the DKK1 of HD patients in this study. These findings indicated that sclerostin might play a role as well as be a biomarker of osteoporosis in HD patients.

## Figures and Tables

**Figure 1 medicina-58-00385-f001:**
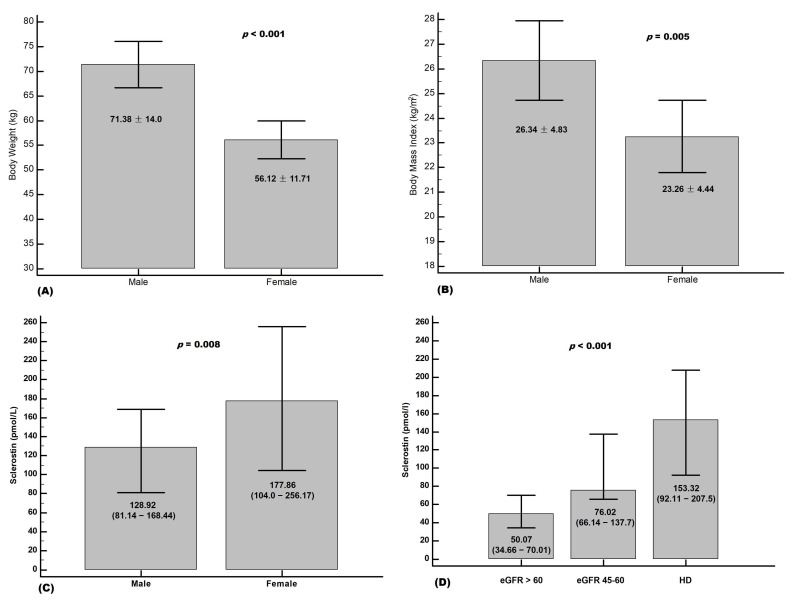
Difference of (**A**) body weight, (**B**) body mass index, (**C**) sclerostin between male and female HD patients, and (**D**) levels of sclerostin among HD patients, eGFR more than 60 min/min/1.73 m^2^ and 45–60 mL/min/1.73 m^2^.

**Table 1 medicina-58-00385-t001:** Clinical characteristics of hemodialysis patients and control group.

Characteristics	HD Patients (*n* = 75)	Control Group (*n* = 65)	*p* Value
Lumbar BMD (g/cm^2^)	0.95 ± 0.19	0.91 ± 0.18	0.288
Lumbar T-score	−0.58 ± 1.55	−0.59 ± 1.68	0.965
Age (years)	66.04 ± 9.08	65.51 ± 4.83	0.225
Female, *n* (%)	38 (50.7)	29 (44.6)	0.475
Waist circumference (cm)	90.93 ± 11.83	83.95 ± 7.83	<0.001 *
Body weight (kg)	63.65 ± 14.94	64.09 ± 9.88	0.839
Body height (cm)	159.79 ± 8.12	161.80 ±8.54	0.155
BMI (kg/m^2^)	24.78 ± 4.86	24.51 ± 3.44	0.714
SBP (mmHg)	139.89 ± 26.26	135.54 ± 11.39	0.197
DBP (mmHg)	73.64 ± 14.00	75.91 ± 5.30	0.220
BUN (mg/dL)	57.55 ± 13.12	17.71 ± 5.11	<0.001 *
Creatinine (mg/dL)	9.11 ± 1.83	1.06 ± 0.20	<0.001 *
Estimated GFR (mL/min/1.73 m^2^)		67.68 ± 17.55	
45–60, *n* (%)		25	
>60, *n* (%)		40	
Alkaline phosphatase (U/L)	79.00 (61.00–107.00)	71.00 (61.5–96.50)	0.189
Total calcium (mg/dL)	8.96 ± 0.73	9.56 ± 0.90	<0.001 *
Phosphorus (mg/dL)	4.49 ± 1.23	3.27 ± 0.71	<0.001 *
iPTH (pg/mL)	231.53 ± 186.60	43.46 ± 9.85	<0.001 *
Sclerostin (pmol/L)	153.32 (92.11–207.50)	63.37 (40.52–83.27)	< 0.001 *
Dickkopf-1 (pmol/L)	13.63 (7.42–21.87)	7.00 (4.88–10.03)	< 0.001 *
Diabetes mellitus, *n* (%)	34 (45.3)	24 (36.9)	0.314
Hypertension, *n* (%)	33 (44.0)	27 (41.5)	0.769

Values for continuous variables are shown as mean ± standard deviation after analysis by Student’s *t*-test; variables not normally distributed are shown as median and interquartile range after analysis by the Mann–Whitney U test; values are presented as number (%) and analysis after analysis by the chi-square test. * *p* < 0.05 was considered statistically significant; BMD, body mineral density; HD, hemodialysis; BMI, body mass index; SBP, systolic blood pressure; DBP, diastolic blood pressure; BUN, blood urea nitrogen; GFR, glomerular filtration rate; iPTH, intact parathyroid hormone.

**Table 2 medicina-58-00385-t002:** Clinical characteristics by different lumbar T-score (normal, osteopenia, and osteoporosis) of the hemodialysis patients.

Characteristics	Normal	Osteopenia	Osteoporosis	*p* Value
Patient number	45	20	10	
Lumbar BMD (g/cm^2^)	1.07 ± 0.13	0.82 ± 0.06	0.65 ± 0.05	< 0.001 *
Lumbar T-score	0.42 ± 1.04	−1.64 ± 0.43	−2.96 ± 0.41	< 0.001 *
Age (years)	63.47 ± 8.87	69.20 ± 8.22	71.30 ± 8.03	0.008 *
Female, n (%)	15 (33.3)	14 (70.0)	9 (90.0)	0.001 *
HD duration (months)	55.44 (20.22–102.60)	58.86 (18.63–115.95)	41.58 (21.27–164.61)	0.842
Waist circumference (cm)	94.51 ± 11.74	88.40 ± 8.62	79.90 ± 10.27	0.001 *
Body weight (kg)	68.89 ± 14.85	60.19 ± 10.68	46.96 ± 6.50	< 0.001 *
Body height (cm)	162.07 ±7.41	159.00 ± 6.45	151.10 ± 8.63	< 0.001 *
BMI (kg/m^2^)	26.17 ± 5.09	23.76 ± 3.86	20.55 ± 2.09	0.002 *
SBP (mmHg)	143.69 ± 24.35	136.50 ± 29.06	129.60 ± 27.78	0.248
DBP (mmHg)	75.16 ± 14.94	73.30 ± 12.28	67.50 ± 12.13	0.296
BUN (mg/dL)	56.44 ± 12.63	57.15 ± 14.74	63.30 ± 11.50	0.327
Creatinine (mg/dL)	9.47 ± 1.79	8.79 ± 1.82	8.12 ± 1.68	0.069
Alkaline phosphatase (U/L)	73.00 (60.00–97.25)	86.00 (68.00–125.5)	99.50 (79.0–127.0)	0.025 *
Total calcium (mg/dL)	8.92 ± 0.68	9.14 ± 0.79	8.78 ± 0.84	0.385
Phosphorus (mg/dL)	4.67 ± 1.09	4.29 ± 1.40	4.12 ± 1.44	0.307
iPTH (pg/mL)	205.70 ± 178.57	275.15 ± 195.76	257.96 ± 202.25	0.348
Urea reduction rate	0.72 ± 0.04	0.74 ± 0.05	0.77 ± 0.04	0.006 *
Kt/V (Gotch)	1.29 ± 0.15	1.37 ± 0.20	1.46 ± 0.15	0.008 *
Sclerostin (pmol/L)	122.04 (72.85–168.44)	169.00 (127.85–221.54)	390.73 (373.11–410.38)	< 0.001 *
Dickkopf-1 (pmol/L)	9.85 (7.30–20.64)	19.27 (8.65–26.78)	14.99 (8.56–20.36)	0.413
Diabetes mellitus, *n* (%)	23 (51.1)	7 (35.0)	4 (40.0)	0.453
Hypertension, *n* (%)	20 (44.4)	10 (50.0)	3 (30.0)	0.579

Values for continuous variables given as means ± standard deviation and test by one-way analysis of variance; variables not normally distributed given as medians and interquartile range and test by Kruskal–Wallis analysis. * *p* < 0.05 was considered statistically significant after Kruskal–Wallis analysis or one-way analysis of variance. Kt/V, fractional clearance index for urea; HD, hemodialysis; BMI, body mass index; SBP, systolic blood pressure; DBP, diastolic blood pressure; BUN, blood urea nitrogen; iPTH, intact parathyroid hormone.

**Table 3 medicina-58-00385-t003:** Correlation of lumbar bone mineral density levels and clinical variables.

Variables	Lumbar BMD (g/cm^2^)				
	Univariate		Multivariate		
	*r*	*p* Value	Beta	Adjusted *R^2^* Change	*p* Value
Female	−0.337	0.003 *	−0.288	0.072	0.001 *
Diabetes mellitus	0.040	0.730	-	-	-
Hypertension	0.055	0.641	-	-	-
Age (years)	−0.337	0.003 *	−0.216	0.041	0.007 *
Log-HD duration (months)	−0.022	0.852	-	-	-
Body height (cm)	0.495	<0.001 *	-	-	-
Body weight (kg)	0.545	<0.001 *	-	-	-
Waist circumference (cm)	0.397	<0.001 *	-	-	-
BMI (kg/m^2^)	0.406	<0.001 *	-	-	-
SBP (mmHg)	0.145	0.216	-	-	-
DBP (mmHg)	0.128	0.275	-	-	-
BUN (mg/dL)	−0.053	0.650	-	-	-
Creatinine (mg/dL)	0.277	0.016 *	-	-	-
Alkaline phosphatase (U/L)	−0.278	0.016 *	-	-	-
Total calcium (mg/dL)	−0.090	0.444	-	-	-
Phosphorus (mg/dL)	0.153	0.190	-	-	-
iPTH (pg/mL)	−0.200	0.088	-	-	-
Sclerostin (pmol/L)	−0.679	<0.001 *	−0.546	0.454	<0.001 *
Dickkopf-1 (pmol/L)	0.112	0.337	-	-	-
Urea reduction rate	−0.366	0.001 *	-	-	-
Kt/V (Gotch)	−0.347	0.002 *	-	-	-

Data of HD duration showed skewed distribution and therefore were log-transformed before analysis. Kt/V, fractional clearance index for urea; HD, hemodialysis; BMI, body mass index; SBP, systolic blood pressure; DBP, diastolic blood pressure; BUN, blood urea nitrogen; iPTH, intact parathyroid hormone * *p* < 0.05 was considered statistically significant.

**Table 4 medicina-58-00385-t004:** Correlation of sclerostin levels and clinical variables by simple or multivariable linear analyses of the hemodialysis patients.

Variables	Sclerostin (pmol/L)				
	HD Patients				
	Simple Regression		Multivariable Regression		
	*r*	*p* Value	Beta	Adjusted *R^2^* Change	*p* Value
Lumbar BMD (g/cm^2^)	−0.679	< 0.001 *	−0.687	0.465	< 0.001 *
Lumbar T-score	−0.687	< 0.001 *	—	—	—
Age (years)	0.175	0.134	—	—	—
Female, n (%)	0.331	0.004 *	—	—	—
Log-HD duration (months)	0.053	0.649	—	—	—
Waist circumference (cm)	−0.394	< 0.001 *	—	—	—
Body weight (kg)	−0.491	< 0.001 *	—	—	—
Body height (cm)	−0.443	< 0.001 *	—	—	—
BMI (kg/m^2^)	−0.382	0.001 *	—	—	—
SBP (mmHg)	−0.092	0.431	—	—	—
DBP (mmHg)	−0.025	0.816	—	—	—
BUN (mg/dL)	−0.027	0.818	—	—	—
Creatinine (mg/dL)	−0.212	0.068	—	—	—
Alkaline phosphatase (U/L)	0.238	0.039 *	—	—	—
Total calcium (mg/dL)	−0.094	0.423	—	—	—
Phosphorus (mg/dL)	−0.219	0.059	—	—	—
iPTH (pg/mL)	−0.0004	0.997	—	—	—
Urea reduction rate	0.346	0.002 *	—	—	—
Kt/V (Gotch)	0.329	0.004 *	—	—	—
Dickkopf-1 (pmol/L)	0.022	0.848	—	—	—

Data of HD duration showed skewed distribution and therefore were log-transformed before analysis. Analysis of data was done using the simple linear regression analyses or multivariable stepwise linear regression analysis (adapted factors were lumbar BMD, lumbar T-score, female, waist circumference, body weight, body height, body mass index, alkaline phosphatase, Urea reduction rate, and Kt/V). BMD, body mineral density; SBP, systolic blood pressure; DBP, diastolic blood pressure; HD, hemodialysis; iPTH, intact parathyroid hormone; Kt/V, fractional clearance index for urea. * *p* < 0.05 was considered statistically significant.

## Data Availability

The data presented in this study are available on request from the corresponding author.

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
