# Peer review of "Serum Sclerostin Level Is Negatively Associated with Bone Mineral Density in Hemodialysis Patients"

_medicina, 2022, doi:10.3390/medicina58030385_

Round 1

Reviewer 1 Report

A cross-sectional study revealing the relationship between DKK1 and sclerostin and BMD in a relatively small number of hemodialysis patients.Sclerostin was negatively correlated with BMD, but DKK1 was not significantly associated with BMD.

The authors clarified the relationship between sclerostin and BMD and the relationship between DKK1 and BMD in hemodialysis patients. It is considered to be a valuable study showing that sclerostin can be a powerful biomarker of bone metabolism in hemodialysis patients.

However, the author should correct the following two points.

  1. The text in "2.3 Biochemical investigations" of Method section is the same as the text in "2.4 Bone mineral density measurements". Perhaps the text for "2.4" was mistakenly written in "2.3". This needs to be fixed.
  2. In Discussion, lines 214-216, the authors write: "this study highlighted a negative correlation between sclerostin and BMD which in accordance with current studies using agents antagonizing sclerostin to prevent osteoporosis." However, it seems that this study did not consider drugs. Please explain this sentence a little more.

Author Response

Comments and Suggestions for Authors

A cross-sectional study revealing the relationship between DKK1 and sclerostin and BMD in a relatively small number of hemodialysis patients. Sclerostin was negatively correlated with BMD, but DKK1 was not significantly associated with BMD.

The authors clarified the relationship between sclerostin and BMD and the relationship between DKK1 and BMD in hemodialysis patients. It is considered to be a valuable study showing that sclerostin can be a powerful biomarker of bone metabolism in hemodialysis patients.

However, the author should correct the following two points.

  1. The text in "2.3 Biochemical investigations" of Method section is the same as the text in "2.4 Bone mineral density measurements". Perhaps the text for "2.4" was mistakenly written in "2.3". This needs to be fixed.

Ans: Thanks for your comments. We re-wrote this paragraph (2.3. Biochemical investigations.)

  1. In Discussion, lines 214-216, the authors write: "this study highlighted a negative correlation between sclerostin and BMD which in accordance with current studies using agents antagonizing sclerostin to prevent osteoporosis." However, it seems that this study did not consider drugs. Please explain this sentence a little more.

Ans: Thanks for your comments. We revised the description of this sentence as “Taken together, although there was literature showing relationship between sclerostin and BMD, the results were controversial. In this study, we showed that levels of sclerostin was higher in female than male; positively associated with aging but negatively associated with BMI and body weight as well as lumbar BMD. The results of this study were in accordance with previous studies using agents antagonizing the effects of sclerostin to increased BMD and to prevent osteoporosis in post-menopause women [31-33], and there was similar finding with a negative association between sclerostin and BMD in HD patients”

Reviewer 2 Report

Dear Authors,
I read this paper investigating the association of BMD in HD patients with clinical and biochemical parameters.
Studies on bone metabolism and health in HD patients are always interesting, but I think that the authors should discuss the limited novelty of this study since the results presented here are confirmative of data already available in the literature. 
Moreover, I have some other remarks:
-      In the Abstract, the authors report that “sclerostin level, and percentage of female patients significantly positively associated with… body mass index”, which is apparently in contrast with the following results reported. Please, explain or correct this sentence
-      Section 2.2 and 2.3 are duplicate, while the section “Biochemical investigations“completely lacks
-      The authors should declare the number of patients involved at the beginning of the result section
-      Did the authors compare HD patients with control subjects? They should provide some information on the number and enrollment strategy of controls, and they should add a table reporting the comparisons between HD and controls (including statistical significance) 
-      There are any data on vitamin D treatment?
-      Since the authors focused their attention on sclerostin, I think that it is necessary to evaluate the association of serum sclerostin levels with other biochemical factors involved in CKD-MBD, such as PTH, phosphate, and vitamin D levels (for example, see Behets GJ, Circulating levels of sclerostin but not DKK1 associate with laboratory parameters of CKD-MBD. PLoS One. 2017 May 11;12(5):e0176411).

Author Response

Comments and Suggestions for Authors

Dear Authors,
I read this paper investigating the association of BMD in HD patients with clinical and biochemical parameters.
Studies on bone metabolism and health in HD patients are always interesting, but I think that the authors should discuss the limited novelty of this study since the results presented here are confirmative of data already available in the literature. 

Ans: Thanks for your comments. We revised the discussion as “Taken together, although there was literature showing relationship between sclerostin and BMD, the results were controversial. In this study, we showed that levels of sclerostin was higher in female than male; positively associated with aging but negatively associated with BMI and body weight as well as lumbar BMD. The results of this study were in accordance with previous studies using agents antagonizing the effects of sclerostin to increased BMD and to prevent osteoporosis in post-menopause women [31-33], and there was similar findings with a negative association between sclerostin and BMD in HD patients”.

Moreover, I have some other remarks:
-      In the Abstract, the authors report that “sclerostin level, and percentage of female patients significantly positively associated with… body mass index”, which is apparently in contrast with the following results reported. Please, explain or correct this sentence

Ans: Thanks for your comments. We correct the description of abstract as “Age, alkaline phosphatase, urea reduction rate, fractional clearance index for urea, sclerostin level, and percentage of female patients significantly negatively associated with, while the body mass index and waist circumference significantly positively associated with the lumbar BMD and T-score.”

-      Section 2.2 and 2.3 are duplicate, while the section “Biochemical investigations“completely lacks

Ans: Thanks for your comments. We re-wrote this paragraph (2.3. Biochemical investigations.).

-      The authors should declare the number of patients involved at the beginning of the result section
-      Did the authors compare HD patients with control subjects? They should provide some information on the number and enrollment strategy of controls, and they should add a table reporting the comparisons between HD and controls (including statistical significance) 

Ans: Thanks for your comments. We revised the manuscript in the Results section as “In this study, there were 75 HD and 65 age-, gender- and DEXA-matched control participants enrolled.” And presented this results as Table 1.

-      There are any data on vitamin D treatment?

Ans: Thanks for your comments. In this study, we did not measure serum vitamin D levels and medications used of HD patients did not record. We would discuss this point in the Limitation section.

-      Since the authors focused their attention on sclerostin, I think that it is necessary to evaluate the association of serum sclerostin levels with other biochemical factors involved in CKD-MBD, such as PTH, phosphate, and vitamin D levels (for example, see Behets GJ, Circulating levels of sclerostin but not DKK1 associate with laboratory parameters of CKD-MBD. PLoS One. 2017 May 11;12(5):e0176411).

Ans: Thanks for your comments. We re-analyzed and provided Table 4 in the manuscript.

Round 2

Reviewer 2 Report

Dear authors,

I appreciate your attempts to improve the quality of the paper and I find that many important issues have been addressed.

Nevertheless, I think that the paper should be further improved, especially in the sections you revised that are not so clear.

For example, the sentences "There were 65 age-, gender, and DEXA- matched participants were enrolled as the control group", or "Taken together, although the there was literature showing relationship between sclerostin and BMD, the results were controversial" and their related paragraphs should be reformulated to improve their clarity.

Moreover, please use the same abbreviations, for example, "Body height" or "height"?

Finally, I noticed a duplication in Figure 1 (is it true or does it depend on the format of the revised version of the paper?)

Author Response

Dear authors,

I appreciate your attempts to improve the quality of the paper and I find that many important issues have been addressed.

Nevertheless, I think that the paper should be further improved, especially in the sections you revised that are not so clear.

For example, the sentences "There were 65 age-, gender, and DEXA- matched participants were enrolled as the control group", or "Taken together, although there was literature showing relationship between sclerostin and BMD, the results were controversial" and their related paragraphs should be reformulated to improve their clarity.
Ans: Thanks for your comments. We revised the manuscript, Table 2 (use Kruskal-Wallis analysis for alkaline phosphatase, sclerostin and DKK-1) and Figure 1 as well as re-phrased the description for better understanding.

Moreover, please use the same abbreviations, for example, "Body height" or "height"?
Ans: Thanks for your comments. According to your suggestion, we had revised the manuscript with the same abbreviation “body height”.

Finally, I noticed a duplication in Figure 1 (is it true or does it depend on the format of the revised version of the paper?)
Ans: Thanks for your comments. We thought it could be the format of the revised version of this manuscript. We would provide a clean copy (attached as Supplementary file) and “track-change” of the revised manuscript.
